# Opportunistic Premise Plumbing Pathogens. A Potential Health Risk in Water Mist Systems Used as a Cooling Intervention

**DOI:** 10.3390/pathogens10040462

**Published:** 2021-04-12

**Authors:** Edmore Masaka, Sue Reed, Maggie Davidson, Jacques Oosthuizen

**Affiliations:** Public Health and Occupational Health and Safety, School of Medical and Health Sciences, Edith Cowan University, Joondalup, WA 6027, Australia; s.reed@ecu.edu.au (S.R.); Ma.Davidson@westernsydney.edu.au (M.D.); j.oosthuizen@ecu.edu.au (J.O.)

**Keywords:** water mist systems, opportunistic premise plumbing pathogens, legionella pneumophila, mycobacterium avium, pseudomonas aeruginosa, acanthamoeba, naegleria fowleri

## Abstract

Water mist systems (WMS) are used for evaporative cooling in public areas. The health risks associated with their colonization by opportunistic premise plumbing pathogens (OPPPs) is not well understood. To advance the understanding of the potential health risk of OPPPs in WMS, biofilm, water and bioaerosol samples (n = 90) from ten (10) WMS in Australia were collected and analyzed by culture and polymerase chain reaction (*PCR*) methods to detect the occurrence of five representative OPPPs: *Legionella pneumophila*, *Pseudomonas aeruginosa*, *Mycobacterium avium*, *Naegleria fowleri* and *Acanthamoeba*. *P. aeruginosa* (44%, n = 90) occurred more frequently in samples, followed by *L. pneumophila* serogroup (Sg) 2–14 (18%, n = 90) and *L. pneumophila* Sg 1 (6%, n = 90). A negative correlation between OPPP occurrence and residual free chlorine was observed except with *Acanthamoeba*, *rs* (30) = 0.067, *p* > 0.05. All detected OPPPs were positively correlated with total dissolved solids (TDS) except with *Acanthamoeba*. Biofilms contained higher concentrations of *L. pneumophila* Sg 2–14 (1000–3000 CFU/mL) than water samples (0–100 CFU/mL). This study suggests that WMS can be colonized by OPPPs and are a potential health risk if OPPP contaminated aerosols get released into ambient atmospheres.

## 1. Introduction

Water mist systems (WMS) are premise plumbing installations used for cooling and are typically installed in outdoor areas to produce and release water aerosols that flash evaporate in the surrounding air, resulting in a sudden reduction of ambient temperatures. Premise plumbing refers to all the water distribution and storage infrastructure within buildings and downstream from the water meter. Water mist systems present a potential public health risk because of their shared characteristics with other aerosol generating premise plumbing systems such as cooling towers, spa pools and showers that have been associated with outbreaks of infectious respiratory diseases caused by OPPPs such as Legionnaires’ disease and bacterial pneumonia [1,2]. These systems produce microscopic inhalable aerosols (0.3–10 µm) [3], which if produced from contaminated water sources, can cause debilitating and fatal respiratory infections. Microorganisms that colonize and regrow in these premise plumbing systems are often referred to in the literature as opportunistic premise plumbing pathogens (OPPPs) and are part of the normal microbiome of premise plumbing [4], which includes showers [5], garden hoses [6], water taps and faucets [7], hot water systems [8], spa pools [9] and air conditioning units [10].

Several characteristics common to premise plumbing that can enhance the risk of microbial colonization and proliferation are oligotrophic conditions, water stagnation and long periods of water retention within plumbing systems [11]. Plumbing materials and components, disinfection methods, system corrosion, water quality/source and elevated temperatures are known to influence the survival of these pathogens in premise plumbing [11,12]. Other features that enhance the survival of OPPPs include their ability to form and colonize biofilms, survival inside free-living amoeba (FLA), and resistance to disinfectants [13]. *Acanthamoeba* has a significant ability to engulf other OPPPs, and through this process shields them from disinfectants such as chlorine, and at the same time confer increased virulence to these OPPPs, that are then able to multiply in premise plumbing [13]. Opportunistic pathogens commonly isolated from premise plumbing include *Legionella pneumophila*, *Mycobacterium avium*, *Pseudomonas aeruginosa*, *Acanthamoeba* and *Naegleria fowleri* [14]. These opportunistic pathogens represent an increased public health risk of *L. pneumophila* infection in persons with compromised immunity [15], as well as the elderly and smokers [16].

Exposure to contaminated waters is an important pathway for infection with OPPPs with inhalation, aspiration and nasal irrigation being the major routes of exposure [17]. Various pneumonic and respiratory tract illnesses have resulted from the inhalation of water mists <10 µm contaminated with bacterial pathogens such as *L. pneumophila* [18,19], *M. avium* [20,21], *P. aeruginosa* [22,23] and the aspiration of water contaminated with *N. fowleri* has resulted in a rare but fatal disease called primary amoebic meningoencephalitis (PAM) [24,25], and infection by *Acanthamoeba* has been associated with diseases of the eyes called acanthamoeba keratitis and granulomatous amoebic encephalitis (GAE) [26].

Although a body of knowledge exists on the presence of OPPPs in premise plumbing features such as showers, water taps, hot water systems, etc., no such study has investigated the potential of WMS used for ambient cooling to be colonized by OPPPs. Currently, there is no literature explaining the environmental characteristics that promote the growth and persistence of OPPPs in these systems. In this study, we investigated the potential occurrence of five selected OPPPs in WMS, namely, *L. pneumophila*, *P. aeruginosa*, *M. avium*, *Acanthamoeba* and *N. fowleri,* to determine the health risks associated with the use of such systems, and to determine whether there is any correlation between the occurrence of the OPPPs in the WMS with residual disinfection, water temperature, water pH, TDS and total organic carbon (TOC).

## 2. Results

### 2.1. Occurrence of Opportunistic Premise Plumbing Pathogens in Water Mist Systems

To determine the occurrence of OPPPs in WMS, we collected 30 bioaerosol samples, 30 biofilm samples and 30 water samples from 10 WMS located in north western Australia. The samples were collected over three sampling events (February, May, and August) during 2019, representing the three climatic seasons of this region. These three seasons are summer, autumn and winter. During summer and the beginning of autumn, daily average temperatures go above 30 °C, often exceeding 35 °C for 6 months of the year, from October to March [27,28]. During the winter months, May–August, average temperatures are often above 20 °C. The annual rainfall rarely exceeds 350 mm [27,28]. These conditions are characterized by a higher rate of evaporation and are ideal for the growth of OPPPs. Both culture and molecular (PCR) methods were used to detect the presence of five representative OPPPs in the samples, namely *L. pneumophila*, *P. aeruginosa*, *M. avium*, *Acanthamoeba* and *N. fowleri*. The water profile parameters of free chlorine residual, temperature, pH, TDS and TOC were also measured and analyzed to determine their relationship with OPPP occurrence in the WMS. Figure 1 shows the frequency of OPPP occurrence in all WMS samples (bioaerosol, water and biofilm). A total of 64 (71%) of WMS samples analyzed tested positive for OPPPs, with *P. aeruginosa* being found in 40 (44%) of the total samples. *L. pneumophila* Sg 2–14 was detected in 16 (18%) of the total samples and *L. pneumophila* Sg 1 was isolated from 5 (6%) of the total samples. Only three of the total samples analyzed returned a positive reading for *Acanthamoeba*. None of the 90 samples analyzed tested positive for both *M. avium* and *N. fowleri.*

### 2.2. The Concentration of Detected OPPPs

The results of this study, as presented in Table 1, show that the concentration of all the OPPPs detected in WMS samples analyzed by microbiological culture methods was higher in biofilm samples than in water samples, with *L. pneumophila* Sg 1 detection in biofilms being 30× higher than in water. The biofilm concentration of *L. pneumophila* Sg 2–14 was three times higher than that of water and *P. aeruginosa* in biofilm samples was eight times higher than in water. The PCR results indicated the presence of *P. aeruginosa* in the bioaerosols only.

### 2.3. The Frequency and Distribution of OPPPs Differed by Sample Type and Water Source

The frequency and distribution of OPPPs differed by the WMS sample type and water source as shown in Figure 1. Bioaerosol samples had a higher occurrence of *P. aeruginosa* (67%) than water samples (40%), and biofilm samples (70%). This occurrence of *P. aeruginosa* significantly differed by sample type χ^2^ (2, N = 90) = 10.08, *p* < 0.05. Conversely, *L. pneumophila* Sg 2–14 occurred more frequently in water samples (37%), than in biofilm samples (17%), however, this difference was not statistically significant χ^2^ (2, N = 90) = 3.07, *p* < 0.05. There was no association between the occurrence of *L. pneumophila* species and *P. aeruginosa* in biofilms and water samples χ^2^ (1, N = 41) = 0.02, *p* > 0.05. V = 0.000. No *L. pneumophila* Sg 2–14 was detected in the bioaerosol samples. Only three biofilm and two water samples tested positive for *L. pneumophila* Sg 1. *Acanthamoeba* was detected in three biofilm samples. *M. avium* and *N. fowleri* were not detected in any of the samples analyzed.

### 2.4. Opportunistic Premise Plumbing Pathogen Occurrence by Water Source

The percentage occurrence of *L. pneumophila* Sg 2–14 in bore water samples as shown in Figure 1 was four times higher than in scheme water; however, the results of a Kruskal-Wallis mean ranks test of the individual occurrences showed that they did not differ significantly, *H (1)* = 1.84, *p* > 0.05. *L. pneumophila* Sg 1 was only detected in five bore water samples.

The results of a Kruskal-Wallis mean ranks test showed a significantly higher percentage occurrence of *P. aeruginosa* in bore water than in scheme water, *H (1)* = 13.87, *p* < 0.05. *Acanthamoeba* was detected in only 2 out of the 36 water samples obtained from systems fed with scheme water and in only one of the water samples obtained from systems fed with bore water.

### 2.5. Seasonal Occurrence of Opportunistic Premise Plumbing Pathogens

In this study, seasonal differences in the occurrence of OPPPs in all samples (N = 90) was investigated, however, no statistical difference was observed in the occurrence of *L. pneumophila* Sg 1, *L. pneumophila* Sg 2–14 and *P. aeruginosa* in WMS across the three seasonal sampling periods (February, May, and August) as indicated by the following results of a Kruskal- Wallis mean rank test for the three OPPPs: *L. pneumophila* Sg 1, *H (2)* = 0.77, *p* = 0.68; *L. pneumophila* Sg 2–14, *H (2)* = 0.89, *p* = 0.64 and *P. aeruginosa*, *H (2)* = 0.08, *p* = 0.96.

### 2.6. Water Temperature

Temperature for all water samples ranged between 21.7 °C to 38.9 °C with the highest being recorded in February and the minimum in May. The results of a Kruskal-Wallis test showed that the mean ranks of water temperature in February were significantly higher than in May and August/September *H (2)* = 23, *p* < 0.05. Based on the results of this study, the occurrence of *P. aeruginosa* in WMS tends to increase with an increase in the water temperature *rs* = 0.31, *p* < 0.05. No correlation was observed between water temperatures and the occurrence of all other OPPPs detected in the WMS namely, *L. pneumophila* Sg 1 *rs* = 0.08, *p* > 0.05, *L. pneumophila* Sg 2–14 *rs* = 0.09, *p* > 0.05 and *Acanthamoeba rs* = 0.04, *p* > 0.05.

### 2.7. Water pH

The pH for all the water samples showed a small range variation (7–7.9). There was no significant difference in the mean ranks of water pH across the three sampling sessions *H (2) =* 0.87, *p* > 0.05.

### 2.8. Total Dissolved Solids (TDS)

The highest TDS concentration was 399 mg/L and was recorded from a bore water sample during the May sampling event. The lowest concentration of 240 mg/L was measured from a scheme water sample during the first sampling event in February. The mean rank concentration of TDS in bore water samples was 6% (18.6 mg/L) higher than in scheme water (340.3 mg/L). This difference was statistically significant *H (1)* = 16.78, *p* < 0.05. No significant difference was noted for the mean ranks of TDS concentration across the three sampling events *H (2)* = 5.33, *p* = 0.07.

### 2.9. Free Chlorine Residual

The concentration of free chlorine residual measured across the three sampling events ranged from 0.0 to 0.76 mg/L, a variance that reflects the complexity of these plumbing systems. The maximum concentration of free chlorine was measured in scheme water during August, with the minimum concentration in this water supply being 0.01 mg/L. Two-thirds of all bore water samples tested across the three sampling events had no free chlorine residual. All scheme water samples tested positive for free chlorine residual. This difference in free chlorine residual between bore and scheme water samples was significant, *H (1)* = 19.95, *p* < 0.05. No significant difference in residual chlorine concentration was observed in the water samples across the three sampling events *H (2)* = 0.26, *p* = 0.88.

### 2.10. Total Organic Carbon (TOC)

Seventy percent (21 out of 30) of the water samples had TOC concentrations less than the detection limit of <1 mg/L and 17 of these were collected from the scheme water supply. The highest measured TOC concentration was 3 mg/L. The mean ranks of TOC concentration in the water samples collected across the seasons were not significantly different *H (2)* = 3.5, *p* = 0.17. However, the TOC concentration in the bore water samples was significantly higher than in the scheme water samples, *H (1)* = 7.11, *p* = 0.01.

### 2.11. The Relationship between Water Profile Parameters

To determine the strength and direction of the association between the water profile parameters discussed above, the nonparametric Spearman’s rho (*rs*) test was used rather than the parametric Pearson test because of the absence of distribution normality in the data sets and the presence of outliers. Table 2 presents the Spearman rho correlation results among the water profile parameters. A significant negative monotonic correlation was determined between free chlorine residual and TDS, *rs* (30) = −0.566, *p* < 0.05 and TOC, *rs* (30) = –0.523, *p* < 0.05. Total organic carbon concentration had a significant and positive monotonic correlation with TDS, *rs* (30) = 0.549, *p* < 0.05. However, there was no significant correlation observed between water temperature and all other water profile parameters, and the same applied to water pH.

### 2.12. Relationship between Water Profile Parameters and the Occurrence of OPPPs in Water Mist Systems

The possible correlation between the water profile parameters and the occurrence of OPPPs in the WMS was determined using the Spearman *rho* correlation test which has been used in similar studies [27]. The results of this analysis are shown in Table 3. Residual chlorine had a significantly weak and negative monotonic correlation with the occurrence of all OPPPs except with *Acanthamoeba*, *rs* (30) = 0.067, *p* > 0.05.

The occurrence of all OPPPs did not correlate with water temperature except for *P. aeruginosa, rs* (30) *=* 0.31, *p* < 0.05. A weak and positive relationship was also observed between TDS concentration and *L. pneumophila* Sg 1, *rs* (30) = 0.27, *p* < 0.05, *L. pneumophila* Sg 2–14, *rs* (30) = 0.42, *p* < 0.05 and *P. aeruginosa*, *rs* (30) = 0.48, *p* < 0.05. The occurrence of both *L. pneumophila* Sg 1 and Sg 2–14 demonstrated a weak positive relationship with TOC, *rs* (30) = 0.39, *p* < 0.05 and *rs* (30) = 0.39, *p* < 0.05, respectively.

## 3. Discussion

The occurrence of OPPPs in WMS used as a cooling intervention in public places has not been investigated, therefore, little is known about their ability to regrow in these systems and whether water profile parameters of temperature, free chlorine residual concentration, pH, TDS and TOC can influence this occurrence. In this study, culture and molecular analysis of 30 biofilm, 30 water and 30 bioaerosol samples collected from 10 WMS confirmed a percentage occurrence of 44% (n = 90) for *P. aeruginosa,* 18% (n = 90) for *L. pneumophila* Sg 2–14, 6% (n = 90) for *L. pneumophila* Sg 1, 3% (n = 90) for *Acanthamoeba* and zero for *M. avium* and *N. fowleri.* As far as we know, this is the first study to investigate the occurrence of these OPPPs in WMS used as a cooling intervention in public places.

In this study, higher concentrations of all OPPPs were detected in WMS biofilm samples than in water and bioaerosol samples, supporting the argument that biofilms play a significant role in OPPP regrowth and survival in water systems. These results are consistent with other studies [29,30,31,32,33]. In our study, *P. aeruginosa* was detected at higher concentrations in WMS biofilms when compared to all the other detected pathogens, a factor which can be attributed to the pathogen’s known ability to colonize and thrive better in biofilms than in the water phase [34]. In interpreting these results, it is important to note that the actual concentration of the OPPPs detected by culture methods could be even higher due to the possible presence of viable but non culturable organisms (VBNC) that may fail to grow under culture conditions [35]. This phenomenon is particularly relevant for *P. aeruginosa*, an opportunistic pathogen that can be affected into the VBNC state by low temperatures during sample transportation [36].

Another reason for the higher numbers of the OPPPs in the WMS biofilms could be the latter’s ability to shield the former from the effect of the chlorine disinfectant used in these systems. Higher disinfection resistance has been demonstrated for the following OPPPs resident in biofilms; *M. avium*, *P. aeruginosa* [37], *L. pneumophila* [38] and *Acanthamoeba* [39].

In this study, a presence of *P. aeruginosa* (67%, n = 30) was detected in WMS bioaerosol samples, indicating that these systems may present a risk of pneumonic infections caused by the inhalation of *P. aeruginosa* [40], which has been established in a number of other studies [10,18,21]. This high detection of *P. aeruginosa* can be attributed to its ability to adapt and thrive better in various environments, such as the one induced by bioaerosol sampling processes. This finding is consistent with another study of *Pseudomonas* occurrence in premise plumbing [41]. Furthermore, research in laboratory models has demonstrated that *P. aeruginosa* is able to remain airborne for periods greater than 45 min [42], whereas *L. pneumophila* is reported to remain airborne for only 3 min [43] after dispersal. By expressing a mucoid phenotype in air, *Pseudomonas* can withstand desiccation common with bioaerosol sampling using filtration [42]. Therefore, *P. aeruginosa* can exist in higher concentrations in ambient atmospheres making it easier to capture during bioaerosol sampling compared to *L. pneumophila*. Further research to investigate this phenomenon in WMS is needed.

*L. pneumophila* Sg 2–14 and Sg 1 were detected in WMS, confirming that these systems could be a health risk for Legionellosis should water aerosols they release when in operation be contaminated by these pathogens, a finding consistent with other studies [18,44,45]. When analyzing for *L. pneumophila* Sg 2–14, only 18% of samples were positive which is greater than another study (11%) [46] and also higher than the levels of *L. pneumophila* Sg 1 (6%), which were lower than in several other studies [33,44].

In this study, a 3% (n = 90) occurrence of *Acanthamoeba* in WMS water and biofilm samples was detected, with this occurrence being positively correlated with free chlorine residual. The positive detection of *Acanthamoeba* in these WMS presents a health risk as described in several studies [26,29,46,47,48], not only because of its pathogenicity, but for its ability to shield other pathogens such as *L. pneumophila* and *M. avium* from destruction by disinfectants such as chlorine [49].

This study did not detect any *M. avium* nor *N. fowleri* in any samples, water (30), biofilms (30) or bioaerosols (30). Although not isolated in any samples, the potential presence of *M. avium* and *N. fowleri* in WMS cannot be completely ruled out, since studies of similar systems have demonstrated that this pathogen can regrow in premise plumbing [29,49,50]. The low sample volumes collected (250 mL) could have resulted in the extracted gene copies being less than the qPCR method’s limit of detection. Sample volumes of 1 L have previously been used to successfully detect these pathogens from water samples [51,52], hence higher sample volumes may be needed for any future studies.

The occurrence of *L. pneumophila* species, *P. aeruginosa*, and thermophilic amoebic species including *Acanthamoeba* in premise plumbing systems tend to vary with seasons [53]. This study did not show a statistical difference across seasons, a result which could be attributed to a loss in statistical power due to the smaller sample size [54]. The mean water temperature measured in the WMS across the three sampling events (29.9 °C) was optimum for the growth of all the targeted OPPPs and could have influenced this result, a finding that is consistent with another study which investigated the critical factors responsible for OPPP growth in premise plumbing [55].

Our study established a correlation between the occurrence of targeted OPPPs in WMS and the use of bore water, with this relationship being significant for *P. aeruginosa, H (1)* = 13.87, *p* < 0.05. One of the factors that could give rise to elevated levels of *P. aeruginosa* and *L*. *pneumophila* Sg 2–14 in the bore water samples could be the increased levels of iron in the shallow aquifers this water is drawn from [56]. Typically, bore water sources in Northern Australia tend to have a higher level of dissolved minerals such as iron, and can also alter the pH of underground water, resulting in the corrosion of pipework and increased colonization of plumbing systems by iron eating bacteria, a finding that is consistent with several other studies [57,58]. In this study, there was no significant difference in the water pH measured across the three sampling events, a finding that could be attributed to the similarity in the chemistry of the source water, which is a shallow aquifer system influenced by infiltration from surface waters [56].

Although the primary source of all the water used in the WMS is drawn from the same aquifer, our research observed a significant variation in the TDS concentration of bore water and scheme water, *H (1) =* 16.78, *p* < 0.05, a result which is not surprising considering that this parameter is usually higher in ground water sources [59].

The positive relationship between the formed biofilms and occurrence of *L. pneumophila* observed in this study is consistent with other studies [55,60], except for the weak correlation with *Acanthamoeba* which may be due to the possible parasitic colonization of free-living amoeba by *L. pneumophila* at water temperatures > 25 °C [61].

A significant amount of research on OPPP occurrence has demonstrated that elevated water temperatures typical in premise plumbing systems is a critical factor in their survival [11,12,53,62]. However, this study did not demonstrate any correlation between water temperature and the occurrence of all detected OPPPs except with *P. aeruginosa*, *rs* (30) = 0.31, *p* < 0.05, a finding different from several other studies [12,53,62]. The correlation with *P. aeruginosa* occurrence is consistent with existing literature [63], Furthermore, *P. aeruginosa* can adapt to various environmental conditions including surviving temperatures ranging from 10–42 °C and antagonism from other OPPPs [41]. Several reasons could be attributed to this phenomenon, particularly the higher-than-normal annual mean maximum temperatures in the study area that were 32.7 °C in February, 26 °C in May and 29.2 °C in August, time periods that aligned with the three sampling episodes conducted during our study, and with the higher winter temperatures typical of the tropics where this study area is located [28].

Most of the water mist systems are situated outdoors and are reticulated by uninsulated pipework which absorbs elevated levels of radiant heat, resulting in elevated water temperatures that promote the growth of OPPPs as described in a study of temperature variation on OPPPs in domestic plumbing [60]. In interpreting the results of this study, it is important to acknowledge that most of the water temperatures recorded ranged between 21.7 °C to 38.9 °C, a zone known to be optimal for the growth of the detected OPPPs. This meant that assessing the effects of temperature on the detected OPPPs at levels below their optimum growth zone was not possible, considering the tendency of these pathogens to adhere to a threshold related response at temperature extremes [55].

This study established a significant negative correlation between free residual chlorine concentration and the occurrence of most detected OPPPs. This highlights its effectiveness against most OPPPs, except *Acanthamoeba,* a finding consistent with several studies [12,38,64,65,66]. The monochloramine disinfectant used in the WMS is more effective over other forms of chlorine disinfectants because of its longer lasting residual effect, a finding that is consistent with other studies [29,39,67,68]. The positive correlation of residual chlorine and *Acanthamoeba* is consistent with the findings of another study [29]. This could be attributed to several reasons including the possible existence of the cystic form of *Acanthamoeba* detected during our study, which is known to confer resistance to the monochloramine disinfection as previously demonstrated in a previous study [69].

This study determined that the TOC concentration in the WMS water samples was exceptionally low, with 70% (n = 30) being lower than the detection limit of <1 mg/L, although it was positively correlated with the occurrence of *L. pneumophila* Sg 1, *rs* (30) = 0.39, *p* < 0.05 and *L. pneumophila* Sg 2–14, *rs* (30) = 0.39, *p* < 0.05. The low concentration of TOC in WMS is consistent with the findings of several studies of premise plumbing systems that promote the regrowth of these pathogens [55,70].

Several microbiological risk control strategies advocated in guidelines developed to control *Legionella* species in engineered water systems, including evaporative cooling systems, could be applied to WMS because of the similarities that exist between this pathogen and other OPPPs detected in this study. The Health and Safety Executive’s Legionnaire’s disease Technical Guidance HSG 274 Part 2 [71], American National Standard Institute’s ANSI/ASHRE Standard 188–2008 [72] and Australia’s enHealth Guidelines for *Legionella* Control in the operation and maintenance of water systems in health and aged care facilities [73] mandate the implementation of the following control strategies for *Legionella*: risk assessment of water systems for effective design and construction; prevention of water stagnation, implementation of effective maintenance programs and adequate disinfection of water used. These steps avoid the growth of *Legionella* bacteria in these systems, strategies that could be applied to prevent OPPPs growing in WMS investigated in this study.

## 4. Materials and Methods

To determine the health risks associated with the use of WMS as a cooling intervention in public places, a total of 30 water samples, 30 biofilm samples and 30 bioaerosol samples were collected from 10 WMS located in the northwestern part of Australia over three sampling events (February, May, and August) in 2019. For this investigative pilot study, the sample size for each sample type per sampling event was calculated using a confidence level of 95%, population size of 10 WMS and a margin of error of 5%, giving a sample size of 10 per sample type per sampling event. The samples were analyzed at EcoDiagnostic, an Australian laboratory accredited by the National Association of Testing Authorities (NATA).

Ethics approval to conduct this study was obtained from the Edith Cowan University (ECU) Human Research Ethics committee (HREC), Approval Number 16337 MASAKA. Informed consent was obtained from all participants involved in the study.

### 4.1. Bioaerosol Sampling

Bioaerosol samples were collected using the NIOSH BC251–2 stage bioaerosol samplers to which was connected conductive polypropylene filter cassettes loaded with 37 mm polytetrafluoroethylene (PTFE) filters of 3 µm pore size. The sampling was undertaken in accordance with the method described by Coleman, Nguyen [74]. One and half meters of Teflon tubing was used to connect the bioaerosol samplers to SKC AirCheck XR 5000 air sampling pumps that were operated at 3.5 L/minute for a maximum of 30 min to collect positional samples. Before each sampling session, the airflow through the sampler was calibrated, and the flow rate checked after each sampling session, using the SKC Defender 510 Dry Cal standard primary calibrator. Air temperature and humidity was recorded during the sampling process using a Lascar EL-USB-2 humidity and temperature meter and wind speed was also recorded during the sampling process using a Meteos Anemo-Thermometer with a 54 Mm Propeller. The bioaerosol samples were stored and transported on ice at <4 °C to EcoDiagnostic laboratory for analysis using molecular methods for *M. avium, P. aeruginosa*, and *N. fowleri*, *Legionella* species (including *L. pneumophila* Sg 1 and Sg 2–14) and *Acanthamoeba*.

#### Bioaerosol Sample Processing

The inside of the NIOSH BC 25 L, 15 mL and 1.5 mL tubes were rinsed (walls of the tube) with a solution of ATL and proteinase K. The PTFE filters were removed from the cassettes using a filter handling kit and placed inside this solution and vortexed, with a 70% ethanol solution being used to sterilize the forceps after each filter transfer. This solution (with the filter paper) was incubated at 60 °C for 30 min to achieve lysis. Two separate aliquots of this solution (440 µL) were loaded onto the QIAsymphony instrument (QIAGEN) for DNA extraction. The QIAsymphony instrument takes 400 µL of sample and extracts it, eluting into 200 µL. The two extracts were combined and filtered using an AMICON Ultra DNA concentrator, was checked for inhibition at the neat dilution using a PPC qPCR assay and then analyzed neat to detect *M. avium* (qPCR), *Legionella* spp. (PCR), *P. aeruginosa* (qPCR), *Acanthamoeba* (PCR) and *N. fowleri* (qPCR). The qPCR results were expressed qualitatively as detected or not detected. In the absence of a standard method for detecting OPPPs in bioaerosols, validated inhouse PCR and qPCR methods were used as described under analytical methods.

### 4.2. Biofilm Samples

Biofilm samples were collected from the WMS using swabs stored in E-Swab vials containing 1 mL of liquid and sodium thiosulfate to inactivate any residual disinfectants. Swabbing was done following the requirements of the Centers for Disease Control and Prevention (CDC)’s “*Sampling procedure for biofilms in Legionella outbreak investigations”* [75]. The swabbing was done from the inside walls of WMS pipes and sprinkler nozzles. These swabs were put back into the E-Swab vials and transported on ice at 4 °C to EcoDiagnistic laboratory for analysis.

#### Biofilm Swab Sample Preparation

One hundred micro liters (100 µL) of the sample were plated to culture for *Legionella* spp. and *P. aeruginosa* and 100 µL being plated for confirmation. One millilitre (1 mL) each of this preparation was used to culture for *Acanthamoeba* and *N. fowleri* with confirmation being done by PCR. Some samples required dilutions (1:10, 1:100, etc.) to account for the high concentration of background flora. Deoxyribonucleic acid (DNA) was extracted from the swab solution (400 µL and eluted into 200 µL) to detect *M. avium*.

### 4.3. Water Samples

Water samples were collected from the WMS, stored, and transported to the analyzing laboratory following the requirements of “*AS 2013–2012, Water Quality—Sampling for microbiological analysis”* [76]. Sterile plastic bottles (500 mL) treated with sodium thiosulfate to deactivate any available disinfectants were used to collect water samples for microbiological testing for the presence of *L. pneumophila*, *P. aeruginosa*, *M. avium*, *Acanthamoeba* and *N. fowleri*. The bottles were stored and transported on ice at 4 °C to a NATA laboratory for analysis, except for the amoeba samples that were transported at ambient temperature [77]. A calibrated industrial HM Digital TDS and water temperature thermometer with a measuring range of 0–80 °C, and accuracy of ±2%, was used to measure water temperature and total dissolved solids. A Palintest Pooltest 9 Premier water testing unit was used to measure the free chlorine residual disinfectant level, pH and temperature profile of the water samples. 

#### Water Sample Preparation and Analysis

All manipulations associated with sample preparation, culture media, materials and apparatus, enumeration techniques and their selection were conducted as described in “*AS/NZS.1: 2007-Water microbiology: Method 1. General information and procedures (ISO8199:2005, MOD)”* [78]. All samples were handled by trained laboratory staff. *N. fowleri* plates for confirmation were handled in a biosafety cabinet (BSC).

### 4.4. Analytical Methods

#### 4.4.1. Detection and Measurement of *Legionella pneumophila* Species

The detection of *L. pneumophila* in water samples was undertaken according to the requirements of “*AS 3876:2017-Waters-Examination for Legionella spp., including Legionella pneumophila”* [79]. A volume of 0.1 mL of the untreated sample was aseptically inoculated onto 90 mm diameter plates of BCYE and MWY agar and incubated in humid conditions at 32 °C ± 2 °C for 7–10 days. The plates were examined visually on the fourth and last day for *Legionella* colonies that showed iridescence and a change in morphology to granular and similar edges. The presumptive *Legionella* colonies were picked and subcultured onto BCYE and BCYE-Cy agar plates, and incubated in humid conditions at 32 °C ± 2 °C for 3 days. The colonies that grew on the BCYE but failed to do so on the BCYE-Cs were interpreted to be *Legionella* spp.

The confirmation of *L. pneumophila* was performed using a validated inhouse multiplex PCR method (EDP-312). The growing colonies from the BCYE agar plates were lysed in 100 µL of HP water at 95 °C for 5 min to achieve lysis. The purification of the DNA from the prepared isolates was done using the QIAsymphony DNA Mini Kit (192) (QIAGEN) and following the manufacturer’s instructions. The detection of *L. pneumophila* was done by amplifying the following primers and probe sets specific for *ssrA*, *mip* and *wzm*, and based on existing literature [80], Legsp-F (5′-NGG CGA CCT GGC TTC-3′) and Legsp-R (5′-GGT CAT CGT TTG CAT TTA TAT TTA-3′), and Lp-mip-F2 (5′-TTG TCT TAT AGC ATT GGT GCC G-3′) and Lp-mip-R (5′-CCA ATT GAG CGC CAC TCA TAG-3′), and Lp-wzm-F (5′-TGC CTC TGG CTT AGC AGT TA-3′) and Lp-wzm-R (5′-CAC ACA GGC ACA GCA GAA ACA-3′). These primers and probes were used as previously described [80] and were tested for specificity by spiking a sample of pure water with *Legionella* and running a standard PCR and algarose-gel electrophoresis was applied to test for end product specificity. The PCR was then run in a Rotor-Gene Q (QIAGEN) machine following the manufacturer’s instructions under the following cycling conditions: initial denaturation cycle of 1 min at 95 °C followed by 30 cycles for denaturation at 95 °C for 5 s, 30 cycles of annealing at 60 °C for 10 s, extension at 72 °C for 15 s and then an end holding cycle for 7 min at 72 °C. The presence of matching patterns for *L. pneumophila* were observed as follows: PCR fragments 79 bp (10 % tolerance), 110 pb (10% tolerance) and 124 (5% tolerance). The lack of any matching pattern indicated the absence of *Legionella* spp. and the presence of a single matching pattern of 110 pb (10% tolerance) indicated presence of *Legionella* spp. The presence of *L. pneumophila* Sg 2–14 was indicated by 2 matching patterns of 110 pb (10% tolerance) and 124 (5% tolerance) and *L. pneumophila* Sg 1 by all 3 matching patterns.

#### 4.4.2. Detection and Measurement of *Pseudomonas aeruginosa*

The detection and enumeration of *P. aeruginosa* in water samples was done according to the requirements of “*AS/NZS 4276.13.2008 Method 13: Pseudomonas aeruginosa—Membrane filtration method”* [81]. One hundred milliliters (100 mL) of the sample was filtered through a 0.45 µm gridded cellulose acetate membrane filter. The prepared filters containing the filtrate were rolled onto prepared mPA-C agar plates that were then incubated in an inverted position in humid conditions at 41.5 °C ± 0.5 °C for 44 ± 4 hrs with any flat appearing colonies growing on the plates and depicting a light brownish outer rim to the green-black centre recorded as presumptive *P. aeruginosa.*

Confirmation of *P*. *aeruginosa* was determined by a modified and validated qPCR laboratory inhouse method (AS 4276.13 EDP-306). DNA was extracted from the bacterial isolates obtained from the incubated plates using QIAsymphony DNA Mini Kit (192) (QIAGEN) and following the manufacturer’s instructions. The purity of the DNA was achieved by using the commercially available QIAsymphony DNA Kit (QIAGEN) and following the manufacturer’s instructions. *P. aeruginosa* detection was done by amplification in a Roto-Gene Q (QIAGEN) machine and following the manufacturer’s instructions. The following amplicon sequences described in literature [82] were used: forward ETA1: 5′-GAC AAC GCC CTC AGC ATC ACC AGC-3′ and reverse ETA2: 5′-CGC TGG CCC ATT CGC TCC AGC GCT-3′ with a product result of 396 bp. A total volume of 25 µL was used for the PCR. The LightCycler instrument (QIAGEN) was used to achieve the following cycling conditions: 1 denaturation cycle at 95 °C for 3 min, 35 cycles with each one made up of 1 m at 94 °C, 68 °C for 90 s, 72 °C for 1 min and an extension cycle of 10 min at 72 °C.

#### 4.4.3. Detection and Measurement of *Acanthamoeba* and *Naegleria fowleri*

A validated in-house EcoDiagnostics laboratory method (EDP-315), was used to detect and enumerate *Acanthamoeba* and *N. fowleri*. Two hundred and fifty milliliters (250 mL) of the sample, spiked with *E. coli*, were concentrated by centrifugation for both *Acanthamoeba* and *Naegleria* species. The supernatant was poured off, and the pellet was resuspended in the remaining volume. One hundred microliters of the remaining volume were then spread plated onto non nutrient agar (NNA) plate and incubated at 42 °C for 48 h for *Naegleria*, and at 25 °C for 3 days for *Acanthamoeba*, and the presence of amoeba was confirmed using microscopy. Any plaques were picked for confirmation of *Naegleria* sp. by PCR, and then for *N. fowleri* and *Acanthamoeba* by qPCR and PCR, respectively.

For *N. fowleri* confirmation, the cells picked from the incubated NNA agar plates were aseptically transferred into 20 µL of lysis buffer for DNA extraction using the QIAsymphony DNA extraction kit and following the manufacturer’s instructions. The PCR and qPCR were run using the *Naegleria* specific primers and *N. fowleri* specific primers previously described in literature [83,84], respectively. The *Naegleria* spp. PCR amplicon used was sequenced as follows: *Naegleria* spp. forward primer 5′-GAA CCT GCG TAG GGA TCA TTT and reverse primer 5′-TTT CTT TTC CTC CCC TTA TTA-3′ and *N. fowleri* forward primer 5′-GTG AAA ACC TTT TTT CCA TTT-3′ and reverse primer 5′-TTT CTT TTC CTC CCC TTA TTA-3′. The qPCR cycling conditions were: 1 cycle for initial activation at 95 °C for 5 min, followed by 60 cycles for denaturation at 95 °C for 10 s and then 60 cycles for combined annealing and extension at 95 °C for 45 s. Successful PCR amplification was confirmed by the following cycle threshold results in controls; Positive (Ct ≤ 36), Negative (Ct ≥ 37) and NTC control (Ct ≥ 37).

For *Acanthamoeba* confirmation, the twin amplicons JDP1 and JDP2 sequenced respectively as follows: forward primer 5′-GGCCCAGATCGTTTACCGTGAA and reverse primer 5′-TCTCACAAGCTGCTAGGGAGTCA were used for DNA amplification as described in literature [85]. The cycling conditions for *Acanthamoeba* included 1 cycle for initial denaturation at 95 °C for 5 min, followed by 40 cycles for denaturation at 95 °C for 30 s, 40 cycles for annealing at 56 °C for 30 s, 40 cycles for extension at 72 °C for 1 min and then 1 cycle for holding at 72 °C for 7 min. An *Acanthamoeba* PCR amplification was considered successful if the negative control showed no evidence of contamination indicated by the absence of an amplicon band and when the positive control showed a band in line with the expected amplicon of 500 bp ± 25% which was then considered positive for Acanthamoeba and indicated as detected per volume of 250 mL or 1 mL.

#### 4.4.4. Detection and Measurement of *Mycobacteria avium*

The detection of *M. avium* was done using qPCR and *M. avium* specific primers, previously designed and used in literature [86], that target the amplification of the 16S rRNA gene and the IS1311 genetic construct as follows: *Mycobacterium* spp. forward 5′-ATAAGCCTGGGAAACTGGGT-3′ and reverse 5′-CACGCTCACAGTTAAGCCGT3′ with a product target of 484 bp and *M. avium* complex forward 5′-GCGTGAGGCTCTGTGGTGAA-3′ and reverse 5′-ATGACGACCGCTTGGGAGAC-3′ with a product target of 608 bp. One hundred milliliters (100 mL) of the sample were filtered. The resultant filtrate was placed into 2 mL of ATL and ProtK and incubated at 60 °C for 30 min, and then 400 µL was extracted using the QIAsymphony instrument. A 2 µL aliquot of the DNA sample was added to 48 µL of PCR mixture prepared as previously described in literature [86] and ran into a LightCycler 2.0 Machine (QIAGEN) operated according to the manufacturer’s instructions. The following cycling conditions were applied: 1 denaturation cycle at 95 °C for 8 min to achieve activation followed by 29 amplification cycles made up of denaturation for 60 s at 95 °C, annealing for 60 s at 40 °C, extension for 35 s at 72 °C and the last extension cycle for 10 min at 72 °C. A standard PCR and algarose-gel electrophoresis was applied to test for end product specificity.

### 4.5. Data and Statistical Analysis

The continuous water profile data (free chlorine residual concentration, water temperature, water pH, total dissolved solids (TDS) and total organic carbon) was log-transformed and box and whisker plots were used to determine normality before the application of statistical tests. All microbiological culture results for *L. pneumophila* Sg 1, L. *pneumophila* Sg 2–14 and *P. aeruginosa* were reported as colony forming units per milliliter (CFU/mL). The polymerase chain reaction *(PCR)* test results for *M. avium*, *Acanthamoeba*, and *N. fowleri* were reported as detected or not detected and the quantitative polymerase chain *(qPCR)* test results for the bioaerosol samples were reported as detected or not detected.

All sampling results containing censored data reported by the laboratory as being below the detection limits were handled by a non-parametric method advanced by Helsel [85]. Using this method, each of the non-detect values were assigned a value of −1 before the application of the Kruskal-Wallis hypothesis test of significance [87]. This test orders and ranks the data points to indicate the existence of any differences or patterns. This non-parametric test for data sets with non-detects has greater power than parametric tests when the data do not conform to a normal distribution and is preferred over substitution methods that tend to introduce invasive data, often influencing statistical scores [88].

Most of the water profile data were not normally distributed, so the Kruskal-Wallis test of statistical differences between variables (H statistic) was used as an alternative to the one-way analysis of variance (ANOVA). All the OPPP occurrence data was also not normally distributed; therefore, the Spearman rho test and the Chi-square test of association were applied where appropriate to measure the extent of association between water profile variables, and the occurrence of OPPP. Before the application of the Spearman’s rho test, OPPP occurrence data was coded to ‘detected’ where a pathogen had been isolated and ‘not detected’ where the converse was true. The detected and not detected variables were coded to ‘1’ and ‘0’, respectively, to facilitate statistical testing. A significance value of *p* < 0.05 was used to accept or reject the null hypothesis. The Minitab version 18 statistical package was used for all statistical analysis.

## 5. Conclusions

The findings of this study demonstrated that WMS used to cool ambient temperatures are a potential health risk due to colonization by OPPPs such as *L. pneumophila* Sg 1 and Sg 2–14, *P. aeruginosa,* and *Acanthamoeba*, and that factors such as free chlorine residual concentration, TDS concentration and TOC concentration can influence the regrowth of these pathogens in these systems. The current guidelines in Australia, developed partly due to public outrage following isolated outbreaks of *Legionella*, focus more on the control of this pathogen in large facilities such as hospitals, aged care homes and shopping centers, ignoring the health risk posed by other emerging pathogens. Therefore, there is a need to develop guidelines covering a broader range of facilities that may expose people to airborne mists which may contain a range of opportunistic premise plumbing pathogens and review existing public health legislation with the aim of adopting a risk-management approach to ensure the effective control of health risks associated with WMS. Further research is needed to understand the relationship between the water profile in WMS and the survival of OPPPs, and conditions that may result in the release of these pathogens from biofilms and their potential to be released as bioaerosols during aerosolization.

## Figures and Tables

**Figure 1 pathogens-10-00462-f001:**
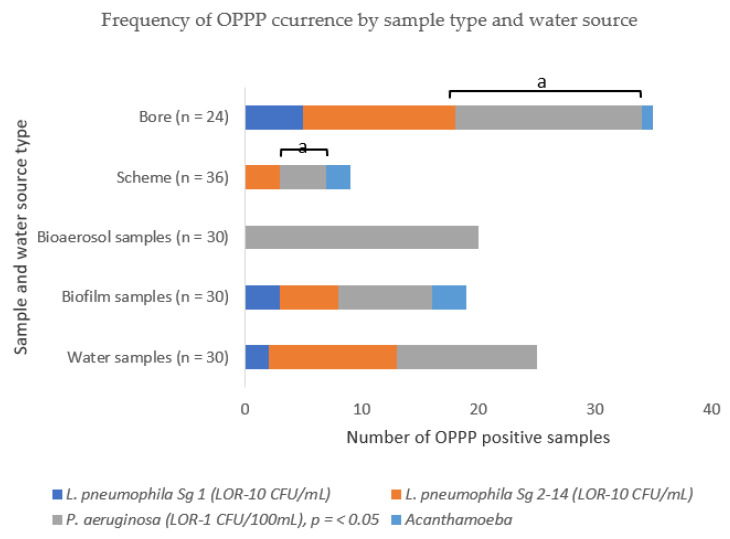
The frequency of OPPP positively identified by sample type and water source. All samples, except bioaerosol samples, were initially identified via culture methods, which were then confirmed via molecular methods similar to the analysis of the bioaerosol samples: PCR/qPCR sensitivities were: *L. pneumophila* −1.6 Genomic Units/mL, *P. aeruginosa* 5–10 GU/10 mL, *Acanthamoeba* 5–8 gene copies/µL. “a” = significant relationship

**Table 1 pathogens-10-00462-t001:** Opportunistic premise plumbing pathogen concentration by sample type.

Opportunistic Pathogen Detected	OPPP Concentration Level	OPPP Concentration Range by Sample Type		
		BIOFILM (CFU/mL)	Water (CFU/mL	Bioaerosol qPCR *
*L. pneumophila (Sg 1)*	Lowest	1000	100	Not detected
	Highest	3000	100	Not detected
*L. pneumophila (Sg 2–14)*	Lowest	100	10	Not detected
	Highest	1000	300	Not detected
*P. aeruginosa*	Lowest	10	3	Detected
	Highest	2000	350	Detected

* PCR and/or qPCR analysis conducted for the detection of OPPPs in bioaerosol samples, results expressed as either detected/not detected.

**Table 2 pathogens-10-00462-t002:** The relationship between water profile parameters.

Spearman Rho (ρ) Correlation between Water Profile Parameters						
Water Profile Parameter	Statistical Test and Sample Size	Free Chlorine Residual	Water Temperature	Water pH	Total Dissolved Solids	Total Organic Carbon
Free chlorine residual	Spearman rho ρ	1	−0.185	−0.065	−0.566	−0.523
	Significance (2 tailed)	.	0.328	0.735	0.001	0.003
	N	30	30	30	30	30
Water temperature	Spearman ρ Correlation	−0.185	1	0.111	−0.089	−0.198
	Significance (2 tailed)	0.328	.	0.558	0.639	0.293
	N	30	30	30	30	30
Water pH	Spearman ρ Correlation	−0.065	0.111	1	0.068	0.279
	Significance (2 tailed)	0.735	0.558	.	0.720	0.136
	N	30	30	30	30	30
Total dissolved solids	Spearman ρ Correlation	−0.566	−0.089	0.068	1	0.549
	Significance (2 tailed)	0.001	0.639	0.720	.	0.002
	N	30	30	30	30	30
Total organic carbon	Spearman ρ Correlation	−0.523	−0.198	0.279	0.549	1
	Significance (2 tailed)	0.003	0.293	0.136	0.002	.
	N	30	30	30	30	30

**Table 3 pathogens-10-00462-t003:** The relationship between water profile parameters and the occurrence of OPPPs in WMS.

Spearman Rho Correlation Analysis between OPPPs and Residual Chlorine, Water Temperature, pH, Total Dissolved Solids, and Total Organic Carbon					
Opportunistic Pathogen Detected	Residual Chlorine (mg/L)	Water Temperature (°C)	Water pH (pH Units)	Total Dissolved Solids (mg/L)	Total Organic Carbon (mg/L)
*L. pneumophila* (1)	−0.327 (*p* = 0.011)	0.080 (*p* = 0.543)	0.074 (*p* = 0.038)	0.268 (*p* = 0.038)	0.392 (*p* = 0.002)
*L. pneumophila* (2–14)	−0.401 (*p* = 0.002)	0.098 (*p* = 0.456)	0.002 (*p* = 0.987)	0.418 (*p* = 0.001)	0.393 (*p* = 0.002)
*P. aeruginosa*	−0.423 (*p* = 0.001)	0.313 (*p* = 0.015)	0.123 (*p* = 0.348)	0.480 (*p* = 0.000)	0.242 (*p* = 0.062)
*Acanthamoeba*	0.067 (*p* = 0.611)	0.035 (*p* = 0.789)	−0.062 (*p* = 0.637)	−0.057 (*p* = 0.663)	0.022 (*p* = 0.868)

## Data Availability

Primer sequences and reaction conditions used in nested PCR amplifications is contained in Appendix A. The primary data used to generate results reported in this study are available on request from the corresponding author subject to applicable restrictions.

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
