# Peer review of "Opportunistic Premise Plumbing Pathogens. A Potential Health Risk in Water Mist Systems Used as a Cooling Intervention"

_pathogens, 2021, doi:10.3390/pathogens10040462_

Round 1
Reviewer 1 Report
“Opportunistic premise plumbing pathogens…” is an interesting work with clear objectives and well written. However, there are some parts of the manuscript that should be modify to improve the quality of the manuscript.
Title, abstract and Introduction are correct.
Results and Material and Methods:
In this work you present the data of opportunistic pathogens from WMS but some points should be clarified.
_ You speak about three climatic seasons in the area studied. Most of the readers of the future of the work live far from Australia and do not know the clime characteristics of the area. So, it would important the description of three seasons in a very short way, especially temperatures and the length of the days. To understand how they can influence the presence of the microbes WMS.
In Figures 1 and 2 and in Table 1 you compare the frequency of detection of the pathogens. However, each one has been detected in a different way and so it is very important to know which is the limit of detection of each OPPP according to the method used and the volume analyzed. From my point of view, this is the weakest part of the work; it is difficult to compare the data when some are obtained with molecular analyses and other by culture. As well, is very difficult to compare the different matrices: mist, water and biofilm. However, the data obtained are interesting.
In table 1 the last column is expressed as qPCR Ct values, I cannot understand this if you use a qPCR. In fact, qPCR shows the quantities of the genomes detected.
In the paragraph 2.2. The concentration… is described in Table 2 instead if Table 1.
In Material and Methods, in 4.4, there are the analytical methods, but you just point out the name of the methods used. I have tried to know how Legionella and Pseudomonas are analyzed according to these regulations, but I have not access to these methods from my computer without paying it. So the methods should be write down summarized, to known how did you analyses the bacteria. As well, the PCR and qPCRs used should be explained in detail: genes amplified temperature, etc.
In 4.4.3 “the protozoa are culture on NNA agar” I understand that NNA is Non Nutrient Agar instead of Nalidixic Agar plate.
Another important point is the interpretation of the possible correlation between OPPP and water characteristics. Legionella and L. pneumophila and Temperature with a 0.08 and 0.09 correlations are not associated with the temperature. As Acanthamoeba is not related with TDS or TOC. Please could you revise these numbers in the table 3 and in the text.
And finally, the discussion should be shorter, it is too long including long dissertations on some points that has not be analyzed in the work,. This happens in most of the paragraphs in page 10.
Author Response
Please see the attached response document to your valuable comments

Reviewer 2 Report
The study under review results interesting and it meets a current issues related to the infection control from waterborne pathogens in community.
Although the Introduction, Results and Discussion sections are well argued and references are updated, some revision are need.
However, I would suggest the authors to organize this paper on all points that I have highlighted.
1) Introduction. Please describe the role of Acanthamoeba in OPPPs spread.
2) Results: Please, ensure that microorganisms names are written in italcs (text and tables)
3) Discussion: This section may be enriched with some information related to the infectious risk management in evaporative coolings (strategies, local guidelines)
4) Methods (bioaresol). Why Legionella and Acanthamoeba were processed with a qualitative PCR?
5) Methods (analytical metohds). Please add some information related to OPPPs isolation methods (culture media, colonies confirmation). Moreover, you may specify all lab modifications. Are Lab methods validated?
Reviewer 3 Report
The use of various devices for the improvement of the environment conditions public places, including water mist systems, should be carefully studied, since such devices are actually bioreactors for opportunistic premise plumbing pathogens (bacteria and protozoa). There are cases when they were a source of human infection and outbreaks of severe infectious diseases. In this regard, the presented work is especially relevant and is of extreme interest. In the research authors investigated the occurrence of selected OPPPs in various environments of WMS to determine the health risks associated with the use water mist systems, and to determine whether there is any correlation between the occurrence of the opportunistic pathogens with environmental parameters, such as residual disinfection, water temperature, water pH, TDS, and TOC.
When reading the manuscript, the following remarks arose:
- For three seasons and three environments, the sample size is small enough. Were adequate statistical methods used, because the distribution could be different from the normal one?
- Figures 1 and 2 should be combined because, in fact, figure 2 details figure 1.
- What is the reason for the different numerical ratios between the content of organisms in biofilms and water, especially for P. аeruginosa? Why does P. aeruginosa appear in aerosols? Is this due to their content in the water of the bores?
- In Figure 3, it is necessary to indicate the bars where the differences are significant.
- Specify in the text - seasonal differences were considered for the entire samples (90) or in some other way?
- It is surprising that there are no statistical differences in the occurrence of opportunistic pathogens by seasons, but at the same time, the occurrence of a number of microorganisms correlates with water temperature, which significantly differs in seasons. This fact requires an explanation.
Round 2
Reviewer 1 Report
Thanks for considering all the comments previously written. The work is much more interesting with all the details added.